# Antibiotic Treatment in *Anopheles coluzzii* Affects Carbon and Nitrogen Metabolism

**DOI:** 10.3390/pathogens9090679

**Published:** 2020-08-21

**Authors:** Estelle Chabanol, Volker Behrends, Ghislaine Prévot, George K. Christophides, Mathilde Gendrin

**Affiliations:** 1Microbiota of Insect Vectors Group, Institut Pasteur de Guyane, 97306 Cayenne, French Guiana; echabanol@pasteur-cayenne.fr; 2Ecole Doctorale Numéro 587, Diversités, Santé, et Développement en Amazonie, Université de Guyane, 97337 Cayenne, French Guiana; 3Tropical Biome and Immunophysiopathology, Université de Guyane, 97300 Cayenne, French Guiana; ghislaine.prevot@univ-guyane.fr; 4Health Sciences Research Centre, University of Roehampton, London SW15 4JD, UK; volker.behrends@roehampton.ac.uk; 5Univ. Lille, CNRS, Inserm, CHU Lille, Institut Pasteur de Lille, U1019—UMR9017—CIIL—Center for Infection and Immunity of Lille, F-59000 Lille, France; 6Department of Life Sciences, Imperial College London, London SW7 2BU, UK; g.christophides@imperial.ac.uk; 7Department of Parasites and Insect Vectors, Institut Pasteur, 75015 Paris, France

**Keywords:** mosquito, microbiota, malaria, metabolism, immunity, tricarboxylic acid cycle, nitrogen excretion, amino acids

## Abstract

The mosquito microbiota reduces the vector competence of *Anopheles* to *Plasmodium* and affects host fitness; it is therefore considered as a potential target to reduce malaria transmission. While immune induction, secretion of antimicrobials and metabolic competition are three typical mechanisms of microbiota-mediated protection against invasive pathogens in mammals, the involvement of metabolic competition or mutualism in mosquito-microbiota and microbiota-*Plasmodium* interactions has not been investigated. Here, we describe a metabolome analysis of the midgut of *Anopheles coluzzii* provided with a sugar-meal or a non-infectious blood-meal, under conventional or antibiotic-treated conditions. We observed that the antibiotic treatment affects the tricarboxylic acid cycle and nitrogen metabolism, notably resulting in decreased abundance of free amino acids. Linking our results with published data, we identified pathways which may participate in microbiota-*Plasmodium* interactions via metabolic interactions or immune modulation and thus would be interesting candidates for future functional studies.

## 1. Introduction

In most mosquito species, females require blood as a source of nutrients for egg production. Blood is assimilated in the midgut (medium intestinal section) and stays stationary for around two days of digestion. As in many other animals, the mosquito midgut is naturally colonized by various microbial communities [1,2,3] composed of fungi, viruses and bacteria, the latter being currently the most studied in mosquitoes. Their bacterial composition is often not very complex, with a few dominant families and genera in most of the individuals [2,4]. However, there can be many variations between different generations of mosquitoes [5] and individuals within a single species [2]. The diversity and composition of this microbiota depend on many complex factors, such as locality [6,7], seasonality [6,8], composition of the water in larval breeding habitats [4,9] and feeding habits of the adults [10]. This composition is also dynamic over the mosquito lifespan, with major changes in crucial stages such as pupation and transformation into adult [7,11]. For females, the blood meal is also a decisive stage in terms of microbial composition, often with a loss of diversity but also an intense proliferation of dominant bacteria [11].

These bacterial communities are not neutral to the mosquito; they interact with their host and affect various aspects of its fitness in a positive or negative way. The microbiota has notably been shown to be essential for larval development as axenic larvae fail to develop beyond the first instar when fed with sterilized conventional food and development can be rescued by adding bacteria into the water [12]. The presence of bacteria seems to have a positive impact on lifespan and fertility in general [13]. However, the intensity of bacterial growth after the blood meal generates a fitness cost to the mosquito, as reduction in this growth positively impacts mosquito lifespan and fertility [14]. Bacteria also contribute to mosquito resistance to insecticides [15] and protect the mosquito from *Plasmodium* infection, hence affecting vector competence [4,5,10].

Indeed, during their blood meal, the female mosquito can incidentally acquire and transmit malaria parasites. If a female *Anopheles coluzzii*, the major malaria vector in Africa, feeds on blood containing infectious forms of *Plasmodium*, the early stages of parasite development will occur in its midgut lumen and epithelium. These stages are key for parasite development as most parasites are lost within the first 24 h, while the surviving ones quickly proliferate for a week at the basal side of the epithelium. Parasites finally move to the salivary glands, colonize saliva and are released by the mosquito during subsequent blood feeding until death [16]. The mosquito midgut is therefore a zone where *Plasmodium* and the microbiota may interact directly or indirectly, via local immune responses.

Some bacteria have been reported to inhibit *Plasmodium* infection, either directly by producing compounds toxic to the parasite [17], or indirectly because they elicit mosquito immunity [5,18,19]. In mammals, metabolic competition is the third mechanism underlying the increased resistance if bacteria-colonized guts to invading pathogens [20]. Whether such competitions are observed between the microbiota and *Plasmodium* and whether such a mechanism also reduces the competence of vector mosquitoes remains an open question.

In this study, we wanted to identify the metabolic pathways that are differentially activated in the presence or absence of bacteria in the gut of sugar-fed and blood-fed mosquitoes that were not infected with *Plasmodium*. To this aim, we analyzed the impact of a high concentration antibiotic treatment from adult emergence on the midgut metabolic profile. Our data provide a glimpse into the effect of the microbiota on the gut metabolism and give some insight into the metabolic interactions that may take place between the mosquito microbiota and *Plasmodium*.

## 2. Results

To investigate whether the microbiota impacts on the mosquito gut metabolome, we treated mosquitoes from emergence with an antibiotic and antifungal cocktail to avoid any colonization of the adult, and dissected mosquitoes either fed only with a fructose solution or 24 h after a blood meal. We first validated the efficiency of this treatment on bacteria in dissected guts, using the bacterial 16S rRNA as a proxy of the bacterial load. We found that the *16S* signal was reduced by 98% and 99% after treatment in sugar-fed and blood-fed mosquito guts, respectively (Figure 1A; *16S*, SF + ABX/SF: *p* = 0.0077; *16S*, BF + ABX/BF: *p* = 0.0040; Tukey test). We found that the genes encoding antimicrobial peptides *CEC1* and *GAM1* were upregulated 4 to 9 fold after blood-feeding in a microbiota-dependent manner (Figure 1B, Appendix A; *CEC1*, BF/SF: *p* = 0.011; *CEC1*, BF + ABX/SF + ABX: *p* = 0.51; *GAM1*, BF/SF: *p* = 0.071; *GAM1*, BF + ABX/SF + ABX: *p* = 0.32; Tukey test). The immune regulator *PGRPLB* was 7-fold induced after blood-feeding, also in a microbiota-dependent manner (BF/SF: *p* = 0.0041; BF + ABX/SF + ABX *p* = 0.67, Tukey test), but was still expressed in antibiotic-treated mosquitoes (Figure 1B). These results are in line with previous reports and indicate that the antibiotic treatment was efficient [18,19,21].

We then analyzed the composition of the gut metabolome in sugar-fed and blood-fed mosquitoes, with and without antibiotic treatment, using samples collected from the same experiments. After methanol extraction followed with gas-chromatography and mass-spectrometry (GC-MS) analysis, we were able to identify and quantify 85 metabolites in our samples. Prior to any normalization, the mean peak intensity of detected metabolites increased 18- to 25-fold after blood-feeding (BF/SF: *p* = 0.0021; BF + ABX/SF + ABX: *p* = 0.0040; *t*-test on geometric means), but was not modified in presence or absence of microbiota (SF + ABX/SF: fold change = 0.79, *p* = 0.52; BF + ABX/BF: fold change = 1.12, *p* = 0.24; *t*-test on geometric means) indicating that the amount of metabolites present inside bacteria is negligible in our analysis. As detailed in the methods, data normalization was then performed in two steps to account for technical variations as well as differences in total amounts of metabolites between conditions.

A principal component analysis (PCA) indicated a clear distinction between sugar-fed and blood-fed samples, matching a first component that explains 48% of the variance, while the influence of the microbiota was less clear and only appeared on the second component when restricting the PCA to blood-fed samples (Figure 2A, Appendix A). The restructuring of the gut metabolome after blood-feeding is further revealed by a significant impact on 48/85 metabolites (Figure 2B, Appendix A; *t*-test without correction, see Methods). Most of these metabolites (79%: 38/48), notably amino acids, were enriched after blood-feeding. Our results are consistent with precedent studies that have compared metabolomic profiles of sugar-fed and blood-fed *Anopheles gambiae* [22,23].

Focusing on the effect of the microbiota, we found that the antibiotic treatment had more reproducible impact on the gut metabolome in blood-fed mosquitoes, where 14 metabolites were significantly affected, than in sugar-fed mosquitoes, where no significant difference on individual metabolites was observed (Figure 3, Appendix A; *t*-test). However, we noted that 11 metabolites were consistently affected with a fold change above 1.7 in at least three replicates in sugar-fed mosquitoes, compared to 5 in blood-fed mosquitoes (Appendix A). This suggests that the impact of the microbiota may be pronounced in sugar-fed conditions as well, but more replication would be required to ascertain it. In blood-fed mosquitoes, we observed that three members of the tricarboxylic acid (TCA) cycle/Krebs cycle were significantly affected: citrate and isocitrate increased after antibiotic treatment, while succinate decreased (citrate: +61%, *p* = 0.032; isocitrate: +54%, *p* = 0.030; succinate: −56%, *p* = 0.011; *t*-test). We did not quantify cis-aconitate, which lies between citrate and isocitrate in the TCA cycle (Figure 4), but its by-products trans-aconitate and itaconate non-significantly accumulated after antibiotic treatment. The accumulation of several metabolites and depletion in succinate suggest that different steps of TCA cycle may happen at different rates, and that the speed of isocitrate to succinate may become limiting. A similar result, albeit non-statistically significant, was observed in sugar-fed mosquitoes.

Our second main observation is that the abundance of several amino acids decreases in blood-fed guts upon antibiotic treatment (Figure 3, Appendix A). This decrease was significant for methionine (−28%, *p* = 0.0013; *t*-test), glutamic acid (−26%, *p* = 0.013; *t*-test), isoleucine (−20%, *p* = 0.0090; *t*-test), and threonine (−12%, *p* = 0.0059; *t*-test), but reflected a more general trend in amino acid abundance. This may be linked to a slower degradation of blood-meal proteins and/or an increased degradation of amino acids. Interestingly, we found that hypoxanthine and uric acid accumulated in antibiotic-treated mosquitoes, albeit significantly for hypoxanthine only (hypoxanthine: +62%, *p* = 0.026; uric acid: +119%, *p* = 0.37; *t*-test). These two metabolites are produced downstream the uric acid cycle, which is responsible for nitrogen excretion in insects. The only two nucleobases that we detected, uracil and adenine, were also present in lower amounts after antibiotic treatment, which was only significant for uracil (uracil: −22%, *p* = 0.046; adenine: −28%, *p* = 0.23; *t*-test). Together, these results suggest that antibiotic treatment leads to a decrease in nitrogen-containing monomers, notably in amino acids and nucleobases, which may be explained by an increase in nitrogen excretion.

## 3. Discussion

In this study, we provide the first analysis of the gut metabolome in sugar-fed and blood-fed *An. coluzzii* under conventional or antibiotic-treated conditions. Our results suggest that the TCA cycle is affected by the antibiotic treatment and reveal a loss of amino acids and other nitrogen-containing metabolites, due to some extent to increased nitrogen excretion.

We first wondered whether these observations are more likely induced by the antibiotic treatment itself or truly reflect microbiota-mediated mechanism. The mechanisms of action of the antimicrobials used in this study are not directly connected to the TCA cycle nor uric acid excretion. Penicillin [24] is a beta-lactam interfering with the biosynthesis of peptidoglycan, a major constituent of bacteria cell wall. Streptomycin [25] and gentamycin [26] bind to 30S-subunit proteins and to 16S rRNA, targeting bacterial translation. Amphotericin B [27] binds to ergosterols in the fungal cell membrane, disrupting membrane integrity. Considering the TCA cycle, we did not find in the literature any evidence for an impact caused by the antibiotics directly on the TCA cycle of mitochondria. However, we cannot rule out any direct impact on mitochondria where this cycle happens. Considering nitrogen excretion, there is evidence that antimicrobials including amphotericin B, gentamycin and streptomycin are nephrotoxic and thus reduce renal function in vertebrates, which can increase uric acid concentration in the serum [28]. This has been reported in mammals, which are ureotelic and in snakes, which as insects are uricotelic (i.e., nitrogen excretion via urea or uric acid, respectively) [29,30,31]. Therefore, the impact we report may (at least partly) be due to similar side-effects, where the accumulation of uric acid and hypoxanthine would result from a blockage in the last steps of excretion. However, transcriptomic data available on Vectorbase [32] suggest that several enzymes of the uric-acid cycle are upregulated in antibiotic-treated mosquitoes [21], pointing that the increase of both metabolites would also be due to their higher production. Further work using gnotobiotic mosquitoes and focusing on diverse tissues (notably Malpighian tubules which are responsible for nitrogen excretion) would allow to further characterize the effect of bacteria and/or antibiotic on nitrogen excretion.

We observed an accumulation in some citrate and isocitrate and a depletion in succinate, suggesting that the rate of the steps of the TCA cycle between isocitrate and succinate became limiting after antibiotic treatment. This may be linked to an increase in glycolysis, suggested by the non-significant decrease in glucose and trehalose, or to a decrease in the isocitrate to succinate conversion itself, for instance if one of its alternative ways is interrupted. Indeed, these steps can be performed either via alpha-ketoglutarate in the conventional TCA cycle or via glyoxylate in the glyoxylate by-pass. The latter is often mentioned to be absent in most Metazoans but is suggested to have appeared via independent horizontal gene transfers between eukaryotes and bacteria and hence to be missed in ortholog-based genome analyses [33].

We found that amino acids are present in higher quantities in conventionally-reared mosquitoes. Besides the suggested increased excretion of free amino acids, antibiotic-treatment may also impact the release of free amino acids via digestion of blood proteins or biosynthesis by bacteria. During blood digestion, amino acids are notably released by bloodmeal-induced carboxypeptidases B [34]. Their role is critical in energy harvest, as shown by the strong decrease in egg production upon treatment of the blood meal by anti-carboxypeptidase antibodies. Interestingly, all four bloodmeal-induced predicted carboxypeptidase genes (ACOM042510, ACOM032536, ACOM024533 and ACOM024534) were found upregulated after antibiotic treatment, 24 h after the blood meal, according to previous transcriptomic data [21,32]. This suggests that amino acid release is not impaired in the presence of antibiotics, but the availability of amino acids may feedback on the expression of these digestive enzymes. Bacteria themselves have also been reported to participate in blood digestion in *Aedes aegypti*, as some antibiotic treatments increased the amounts of non-digested proteins in the mosquito gut, leading to lower egg production [35]. In *Anopheles*, antibiotics do not seem to impair egg production and we even observed that a treatment at a lower concentration increases fecundity [36]. *Drosophila* bacterial symbionts have also been found to secrete several amino acids which are absent from their culture medium [37], notably isoleucine and threonine which are significantly affected by the antibiotic treatment in blood-fed mosquitoes. Altogether, the observed decrease in amino acid levels after antibiotic treatment may be linked to increased nitrogen excretion, decreased digestion efficiency and/or a loss of amino acid secretion by bacteria.

We investigated whether our data can give some hints on the metabolic link between the microbiota and *Plasmodium* in the mosquito. This study did not include any *Plasmodium* infection but antibiotic treatment has repeatedly been found to increase vector competence to *Plasmodium* [5,19,38,39]. We found that free amino acids, glucose and trehalose tended to be present at higher or similar concentrations in the presence of a conventional microbiota to antibiotic-treated mosquitoes. These metabolites have been found to promote *Plasmodium* in mosquito or blood stages [40,41], therefore their decreased abundance in antibiotic-treated mosquitoes indicates that their concentration is still sufficient to support parasite development in antibiotic-treated mosquitoes. On the contrary, we identified three metabolites which may be linked with the positive impact on *Plasmodium* in antibiotic-treated mosquitoes.

First, phosphoenolpyruvate (PEP) is enriched in antibiotic-treated mosquitoes. PEP is thought to be incorporated by PEP carboxylase or PEP carboxykinase into the TCA cycle of *Plasmodium* gametocytes [42]. The importance of PEP at this stage would be consistent with the observation that the apicoplast PEP transporter is required for parasites to go through mosquito stages, while it is dispensable during blood stages.

Secondly, *Plasmodium* is auxotroph for purines; therefore, *Plasmodium falciparum* culture media are typically supplemented with hypoxanthine so that parasites can produce nucleic acids [43,44]. We found that hypoxanthine is enriched in antibiotic-treated mosquitoes, which may be crucial at the beginning of *Plasmodium* development. Indeed, gametogony in the mosquito gut involves DNA replication during the first minutes of gut infection [45] and large amount of DNA are produced in the oocyst stage, where each parasite multiplies to thousands within a week [16].

Thirdly, we observed a depletion in uracil in the gut of antibiotic-treated mosquitoes. In *Drosophila*, the release of uracil in the gut is a signal allowing the immune system to discriminate between commensal and pathogenic gut bacteria. Upon detection of free uracil, the production of reactive oxygen species (ROS) by dual-oxidase is induced, protecting the host against pathogenic infection [46]. The lower amount of uracil may therefore be more favorable for parasitic infections.

In conclusion, we observed that an antibiotic treatment in mosquitoes impacts on carbon and nitrogen metabolism. We identify three candidate metabolites which may impact on vector competence to *Plasmodium*, namely phosphoenolpyruvate, hypoxanthine and uracil. However, more work is required to test these hypotheses and to discriminate between the impact of the microbiota and of the antibiotic treatment.

## 4. Materials and Methods

### 4.1. Mosquito Rearing

The experiments were performed with female *An. coluzzii* from the Ngousso colony (established from field mosquitoes collected in Cameroon in 2006). Larvae were reared in distilled water and fed with TetraMin fish food. Adults were provided 5% (*w*/*v*) fructose ad libitum. Insectary conditions were kept at 27 °C and 80% humidity with a 12 h/12 h light/dark cycle.

### 4.2. Blood Feeding

For experiments, 3- to 6-day-old mosquitoes were fed a first and only blood-meal. Human blood was acquired from the NHS Blood Service using membrane-feeders kept at 37 °C via water circulation. Non-engorged mosquitoes were discarded.

### 4.3. Antibiotic Treatment

The emergence water of pupae was treated with 60 μg/mL Penicillin/Streptomycin (Sigma-Aldrich, Merck, St. Louis, MO, USA), 50 μg/mL Gentamicin (Sigma-Aldrich, Merck, St. Louis, MO, USA) and 0.5 μg/mL Amphotericin B (Fungizone) (Sigma-Aldrich, Merck, St. Louis, MO, USA) and freshly emerged mosquitoes were fed on sterile solution of 5% (*w*/*v*) fructose supplemented with antibiotics at the same concentration. The same treatment was also added to the blood for blood-fed mosquitoes.

### 4.4. Midgut Dissection

Mosquitoes were dissected 24 h after the blood-meal. Before dissection, mosquitoes were immersed in 75% ethanol for 5 min to kill surface bacteria and then washed three times in phosphate-buffered saline solution. Mosquitoes were dissected and 12 to 15 midguts were pooled for each condition for the RT-qPCR analysis (five independent replicates) and 20 midguts were pooled for the metabolomic analysis (four independent replicates, as the first one was used to set up the protocol). The midgut was selected as the portion of the gut located between the cardia (included) and the Malpighian tubules (carefully excluded). Dissections were performed on a slide kept on a mix of ice and dry ice and gut samples were kept in a dry tube (RT-qPCR) or in a tube containing 0.7 mL pre-cooled (<−40 °C) 80% methanol (Metabolomics) in dry ice while dissecting and transferred to −80 °C shortly after.

### 4.5. RNA Extraction and RT-qPCR

Dissected guts were homogenized with beads in TRI Reagent^®^ and chloroform, using Precellys 24 tissue homogenizer (Bertin Technologies, Montigny-le-Bretonneux, France). RNA was precipitated with isopropanol, washed twice in 70% ethanol and resuspended in water. cDNA was synthetized from 325 to 650 ng RNA using Takara Reverse Transcriptase kit. qPCR amplifications were performed in duplicate using the Takara SYBR Premix Ex Taq kit with a total volume of 10 μL on a 7500 Fast Real Time PCR machine (Applied Biosystems, Thermo Fisher Scientific, Waltham, MA, USA). Primer sequences are indicated in Appendix A.

### 4.6. Metabolomics

#### 4.6.1. Methanol Quenching

Freshly dissected samples were homogenized using a Precellys 24 tissue homogenizer (Bertin Technologies, Montigny-le-Bretonneux, France). A negative control with no biological sample was included. Cellular debris were spun down at 14,000 rpm at 4 °C for 15 min and the supernatant was transferred to a fresh Eppendorf tube. The pellet was homogenized and spun again in 0.7 mL of clean pre-cooled 80% methanol, and the second supernatant was added to the first one. Samples were dried in a vacuum drier at 45 °C and kept at −80 °C until dual phase extraction.

#### 4.6.2. Dual Phase Extraction and Sample Preparation

Samples were resuspended in 300 μL of CHCl_3_/MeOH (2:1) and vortexed for 30 s. After addition of 300 μL of water and centrifugation (13,000 rpm, 10 min, room temperature), the top aqueous layer was transferred into an inactivated glass vial and dried before being stored at −80 °C. The lower organic layers were not used in this experiment. For GC-MS, samples were derivatized by a two-step methoximation-silylation derivatization procedure. The dried samples were first methoximated using 20 μL of 20 mg/mL methoxyamine hydrochloride in anhydrous pyridine at 37 °C for 90 min. This was followed by silylation with 80 μL of *N*-methyl-*N*-(trimethylsilyl)trifluoroacetamide (MSTFA) at 37 °C for 30 min [47].

#### 4.6.3. Data Acquisition

GC-MS analysis was performed on an Agilent 7890 Gas Chromatograph (Agilent, Santa Clara, CA, USA) equipped with a 30 m DB-5 ms capillary column with a 10 m DuraGuard column coupled to an Agilent 5975 MSD system (Agilent, Santa Clara, CA, USA) operating under electron impact ionization. Samples were injected with an Agilent 7693 AutoSampler injector (Agilent, Santa Clara, CA, USA) into deactivated spitless liners, following a temperature gradient detailed in Kind et al. 2009 [47], using helium as the carrier gas. Metabolites were identified and quantified using a workflow described by Behrends et al. [48].

### 4.7. Data Analysis

Data were analyzed using the R (version 4.0.0) software and RStudio (version 1.2.5042) software. The ggplot2 package (from Tidyverse) was used for most graphical representations.

#### 4.7.1. RT-qPCR Analysis

For RT-qPCR analysis, relative gene expression ratios were calculated using the Pfaffl method [49] to account for efficiency differences between primers. *RpS7* (ACOM041783) was used as the housekeeping gene and the condition with non-treated sugar-fed mosquitoes was used as a reference for analysis. Average ratios and standard errors were used for graphical representations. Log_10_ transformation was applied to ratios for statistical analysis by ANOVA (lm function in R) followed by Tukey test (TukeyHDS function in R).

#### 4.7.2. Metabolomic Analysis

For metabolomic data, prior to any normalization, geometric means of all metabolite peak intensities were calculated for each sample and conditions were compared with two-sided paired *t*-test. Then, two normalizations were applied to raw data. First, a total peak intensity normalization was performed for each sample, to account for differences in amount of material input. Secondly, a spline-fit normalization was performed with the smoothn function in MATLAB, to correct for metabolite derivatization over time.

The PCA and fviz functions of the FactoMineR and factoextra packages in R were used to perform and plot the PCA, respectively. As mosquitoes taken at the same time point from the colony are more similar between each other, we calculated ratios and operated *t*-test by pairs for analyses and Volcano plots. Paired ratios were calculated and a log_2_ transformation was applied. Threshold was set at log_2_ >1 or <−1 (2-fold enriched) for the BF/SF analysis and was set at >0.3 and <−0.3 (35% enriched) for SF + ABX/SF and BF + ABX/BF analyses. The *p*-values of the two-sided paired *t*-test were examined and metabolites with a *p*-value <0.05 were considered significant. Metabolomic data are very variable; hence, no correction was applied to this *t*-test. This statistical analysis (as any) thus needs to be considered with caution, but this strategy allowed us to better discriminate the metabolites that were most significantly affected compared to those with a high fold change that was not consistent between replicates.

Metabolomic pathways were scrutinized using the Kyoto Encyclopedia of Genes and Genomes (KEGG).

## Figures and Tables

**Figure 1 pathogens-09-00679-f001:**
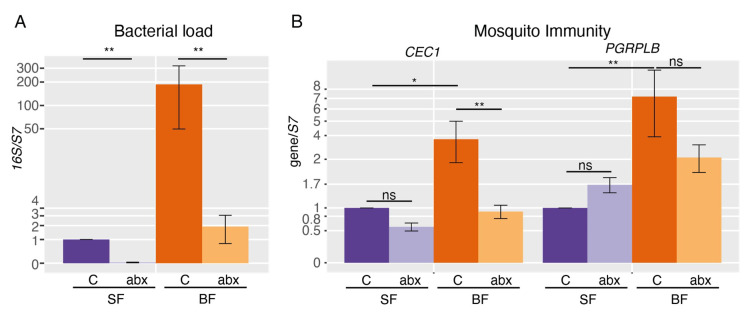
RT-qPCR analysis of bacterial load and immune gene expression. (**A**) Quantification of the bacterial 16S rRNA and (**B**) mosquito immune genes CEC1 and PGRPLB in the gut of sugar-fed (SF) and blood-fed (BF) mosquitoes carrying a conventional microbiota (C) or treated with antibiotics (ABX). Data show the average ratios from 5 independent replicates on a log scale, and error bars represent the standard error. Tukey test was used to determine statistical significance. NS: non-significant, *: *p* < 0.05, **: *p* < 0.01.

**Figure 2 pathogens-09-00679-f002:**
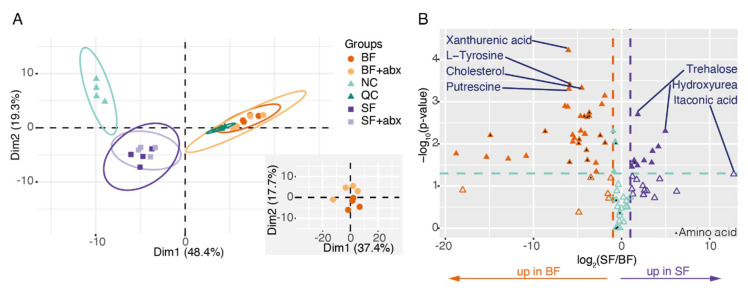
The midgut metabolome is mainly affected by blood versus sugar diet. (**A**) Principal component analysis (PCA) biplots of the metabolomic analysis. Groups are colored by treatment and negative control (NC) and Quality Control (QC) are present. The bottom-right panel represents a PCA focusing on blood-fed samples. (**B**) Volcano plot of the metabolites in sugar-fed versus blood-fed samples. The x axis represents the log_2_ of the ratio of each metabolite between the two conditions. Orange and purple metabolites are >2-fold enriched in blood-fed and sugar-fed mosquito midguts (|log_2_(ratio)| > 1), respectively. The y axis represents the negative log_10_ applied to *p*-values. Metabolites located above the green line are significantly affected (*p* < 0.05; two-sided paired *t*-test; filled triangles) and metabolites below the green line are not (*p* > 0.05; two-sided paired *t*-test; open triangles). Black triangles highlight amino acids.

**Figure 3 pathogens-09-00679-f003:**
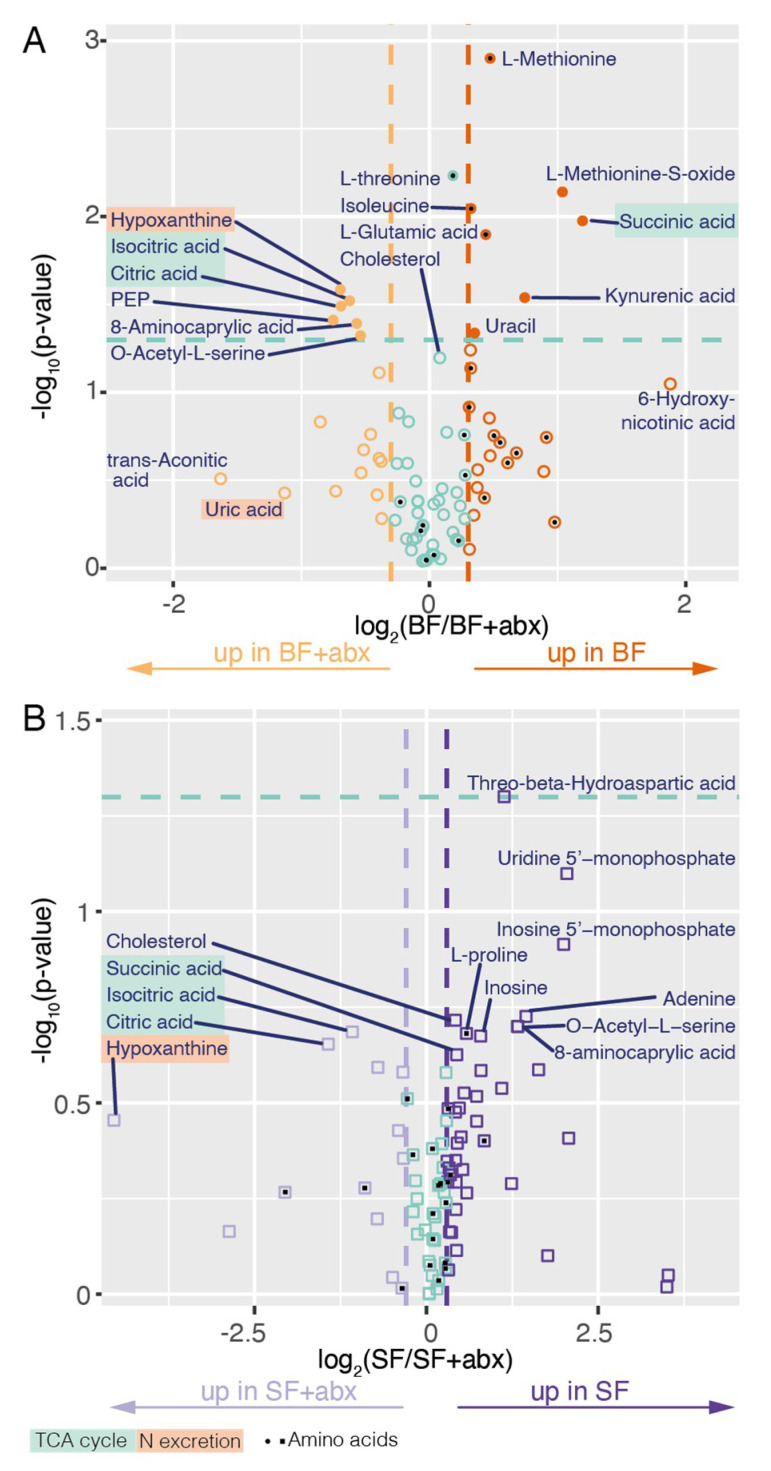
Volcano plot of the metabolites in conventional (BF) versus antibiotic-treated (BF + ABX) blood-fed midguts (**A**) and in conventional (SF) versus antibiotic-treated (SF + ABX) sugar-fed midguts (**B**). The x axis represents the log_2_ of the ratio of each metabolite between the two conditions. Dark and light staining represent metabolites which are increased >35% in conventional and antibiotic-treated samples, respectively (|log_2_(ratio)| > 0.3). The y axis represents the negative log_10_ applied to *p*-values. Metabolites located above the green line are significantly affected (*p* < 0.05; two-sided paired *t*-test; filled symbols) and metabolites below the green line are not (*p* > 0.05; two-sided paired *t*-test; empty symbols). Black squares or circles highlight amino acids.

**Figure 4 pathogens-09-00679-f004:**
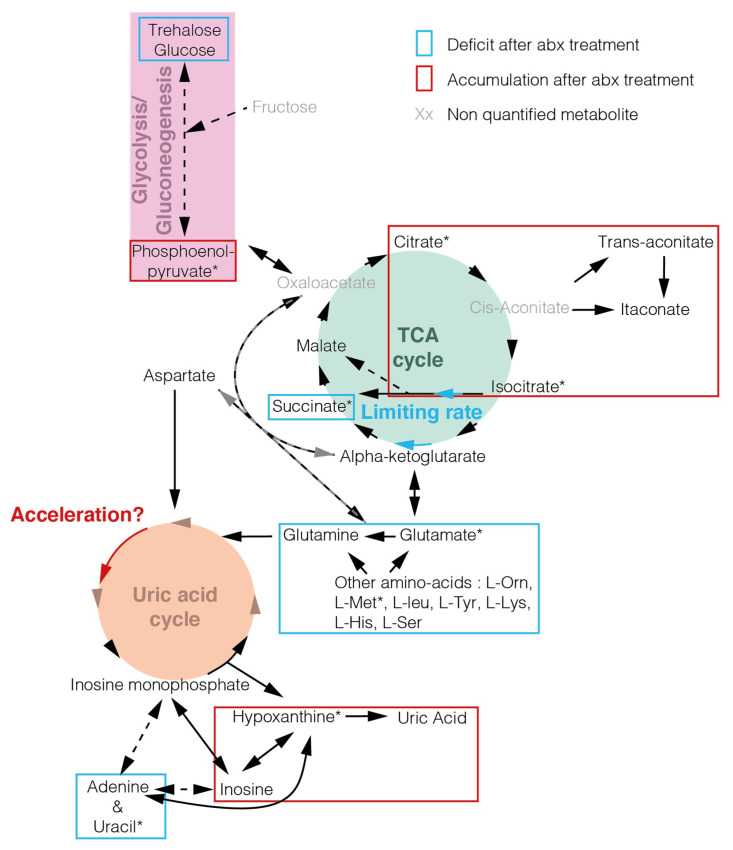
Model—tricarboxylic acid (TCA) cycle and nitrogen metabolism are affected by antibiotic treatment. Metabolites detected in the analysis are shown in black and non-detected metabolites are shown in grey. * highlight significant metabolites between conventional blood-fed mosquitoes and antibiotic-treated ones (two-sided paired *t*-test). Deficits induced by the antibiotic treatment are cycled in blue and accumulations in red. Blue and red arrows indicate potential acceleration and limiting rate, respectively, that are suggested by accumulation or deficit of certain compounds in the pathways. Our data do not allow to determine whether the limiting rate in isocitrate to succinate conversion is linked to a deceleration at this step or to an increase in glycolysis.

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
