# Peer review of "Antibiotic Treatment in Anopheles coluzzii Affects Carbon and Nitrogen Metabolism"

_pathogens, 2020, doi:10.3390/pathogens9090679_

Round 1

Reviewer 1 Report

In the manuscript entitled “Antibiotic treatment in An. coluzzii affects carbon and nitrogen metabolism” Chabanol E. et al. describe the metabolomic differences among adult mid guts of mosquitos feed with a sugar or a blood meal under conventional and antibiotic treated conditions. The authors find no impact of the antibiotic treatment on the mid gut metabolomes of sugar fed animals. In the case of blood fed animals, the authors find 14 metabolites significantly affected by the antibiotic treatment mostly spanning the TCA cycle and amino acids. Using available literature, the authors hypothesize that some of the differentially abundant metabolites may play a role on the microbiota dependent reduction of vector competence to Plasmodium.

The study is interesting and biological question is relevant. The findings on the impact of antibiotic treatment on the metabolic profiles of mid guts from blood feed mosquitos are interesting. The hypothesis related with the microbiota dependent reduction of vector competence to Plasmodium are well founded. However, the lack of conditions including Plasmodium infected mosquitos and gnotobiotic or supplementation experiments make all the conclusions on this matter completely speculative. This should be stated in the abstract.

Line 156: The authors pinpoint the reduction of amino acids in blood feed guts upon antibiotic treatment and hypothesize a slower degradation of proteins or increase of amino acids degradation. Can the amino acids reduction be explained by the lack of bacteria? Do the authors find in the metabolomes an increase in by-products of the gradation of amino acids or small peptides?

The authors do an extended discussion on the impact of the antibiotics on their observations but none on the contribution of gut microbiota on the metabolic profiles identified. Could you develop this on the discussion?

Minor comments:

  • Do not abbreviate organisms name in the tittle
  • Line 33-34: These sentences can be simplified  
  • Line 35: Refs 1-3 do not refer to the co-evolutionary origin of the mosquito symbionts, the text needs to be adjusted
  • Line 47: References are missing
  • Figure 2B: the legend for amino acid is too small
  • Figure 2B, 3A and 3B there is no description of the empty symbols

Author Response

Comment: “the lack of conditions including Plasmodium infected mosquitos and gnotobiotic or supplementation experiments make all the conclusions on this matter completely speculative. This should be stated in the abstract.”

Response: Indeed, infection with Plasmodium is not part of our experiments and further studies including infections need to be conducted to confirm our hypothesis on the microbiota-parasite interactions. We had been careful to not mislead the reader and we pointed it out throughout the text. As suggested, we now made it clearer in the abstract by mentioning that mosquitoes were provided a non-infectious blood-meal (line 22) and by modifying the last sentence of the abstract as such (lines 25-27): “we identified pathways which may participate in microbiota-Plasmodium interactions via metabolic interactions or immune modulation and thus would be interesting candidates for future functional studies.”

Comment: “Line 156: The authors pinpoint the reduction of amino acids in blood feed guts upon antibiotic treatment and hypothesize a slower degradation of proteins or increase of amino acids degradation. Can the amino acids reduction be explained by the lack of bacteria? Do the authors find in the metabolomes an increase in by-products of the gradation of amino acids or small peptides?”

Response: Thank you for this very interesting point, which led us to further investigation. As already stated in the results, our data do not point to a reduction in metabolites due to the simple loss of the metabolites present in bacteria, because the amount of metabolites were similar in presence and absence of bacteria, which are not enriched in amino acids compared to blood. However, we had not considered the secretion of amino acids by bacteria, yet Drosophila symbionts have been found to secrete several amino acids. We now added this point to the discussion (lines 231-237): "Drosophila bacterial symbionts have also been found to secrete several amino acids which are absent from their culture medium [37], notably isoleucine and threonine which are significantly affected by the antibiotic treatment in blood-fed mosquitoes. Altogether, the observed decrease in amino acid levels after antibiotic treatment may be linked to increased nitrogen excretion, decreased digestion efficiency and/or a loss of amino acid secretion by bacteria."

Small peptides were not detected in the GC-MS process and we did not specifically identify degradation by-products.

Comment: “The authors do an extended discussion on the impact of the antibiotics on their observations but none on the contribution of gut microbiota on the metabolic profiles identified. Could you develop this on the discussion?”

Response: As mentioned in the discussion, we realized that at least some part of our results may be linked to direct effects of antibiotics. This likely also applies to other studies based on similar experimental set ups. Therefore, we chose to use a generic terminology “after antibiotic treatment” rather to describe samples as “aseptic” or “germ-free” as found in some articles, in order to keep in mind the presence of xenobiotics in mosquitoes carrying a reduced microbiota. While mentioning “antibiotic treatment”, we also address the role of the microbiota. As this is the first study on the metabolic interactions between the mosquito and its microbiota, this part of the discussion may seem a bit weak. As mentioned above, we now discuss the secretion of amino acids by bacteria isolated from Drosophila and think that this improved the balance of the discussion between microbiota facts and antibiotic effects

Minor Comments: “Do not abbreviate organisms name in the tittle

    Line 33-34: These sentences can be simplified 

    Line 35: Refs 1-3 do not refer to the co-evolutionary origin of the mosquito symbionts, the text needs to be adjusted

    Line 47: References are missing

    Figure 2B: the legend for amino acid is too small

    Figure 2B, 3A and 3B there is no description of the empty symbols”

Response: The abbreviation in the title has been replaced by the full name of Anopheles coluzzii. The first sentences of the introduction have been lightly simplified and the text has been adjusted to match the references. Line 47 is only an introductory sentence and development with according bibliography is made just after in the text. Figures 2 and 3 have been modified for the legends to be more apparent and descriptions of the figures now include the explanation for filled or empty symbols.

We thank the reviewer for his/her valuable suggestions, which helped us to improve the manuscript.

Reviewer 2 Report

This well-written manuscript reports results from a metabolome analysis of the midgut of sugar- and blood-fed Anopheles coluzzi mosquitoes with and without antibiotic treatment. There are several pitfalls associated with the authors' approach, mainly that their use of antibiotic treatment introduces the possibility that some or all of their observations are the result of antibiotic treatment itself and not directly microbiota-mediated. However, these concerns are adequately addressed in the Discussion. 

My only suggestion is that the authors elaborate a bit more on their approach to identify significantly affected metabolites between treatments (described in Lines 320-325). The authors may even consider reporting metabolites that showed high fold changes in response to antibiotic treatment in some but not all replicates as supplementary material.

Author Response

Comment: “My only suggestion is that the authors elaborate a bit more on their approach to identify significantly affected metabolites between treatments (described in Lines 320-325). The authors may even consider reporting metabolites that showed high fold changes in response to antibiotic treatment in some but not all replicates as supplementary material.”

Response: Thanks for these helpful suggestions. Details on the statistical test, calculation of ratios and threshold selection have now been added (lines 345-351): “As mosquitoes taken at the same time point from the colony are more similar between each other, we calculated ratios and operated T-test by pairs for our analysis and for producing the Volcano plots. Paired ratios were calculated and a log2 transformation was applied. Threshold was set at log2 >1 or <-1 (2-fold enriched) for the BF/SF analysis and was set at >0.3 and <-0.3 (20% enriched) for SF+ABX/SF and BF+ABX/BF analyses. P-values of two-sided paired T-test were examined and metabolites with a p-value <0.05 were considered significant.”

We also added in Figure S3 a symbol indicating the metabolites regulated with a fold change over 1.7 in at least three of four replicates. Interestingly, there were more metabolites in sugar-fed mosquitoes than in blood-fed once. Hence, we also modified our initial description of these data in the results (lines 131-137): “Focusing on the effect of the microbiota, we found that the antibiotic treatment had more reproducible impact on the gut metabolome in blood-fed mosquitoes, where 14 metabolites were significantly affected, than in sugar-fed mosquitoes, where no significant difference on individual metabolites was observed (Figure 3, S3; T-test). Yet, we noted that 11 metabolites were consistently affected with a fold change above 1.7 in at least three replicates in sugar-fed mosquitoes, compared to 5 in blood-fed mosquitoes (Figure S3). This suggests that the impact of the microbiota may be pronounced in sugar-fed conditions as well, but more replication would be required to ascertain it.